# Gradient of tactile properties in the rat whisker pad

**Erez Gugig** , **Hariom Sharma**, **Rony Azouz** *

Department of Physiology and Cell Biology, Zlotowski Center for Neuroscience, Ben-Gurion University of the Negev, Beer-Sheva, Israel

☉ These authors contributed equally to this work.
* razouz@bgu.ac.il

**Data Availability Statement:** All relevant data are within the paper and its Supporting Information files. We have attached an Excel file containing all data in the figures.

## Abstract

The array of vibrissae on a rat's face is the first stage in a high-resolution tactile sensing system. Progressing from rostral to caudal in any vibrissae row results in an increase in whisker length and thickness. This may, in turn, provide a systematic map of separate tactile channels governed by the mechanical properties of the whiskers. To examine whether this map is expressed in a location-dependent transformation of tactile signals into whisker vibrations and neuronal responses, we monitored whiskers' movements across various surfaces and edges. We found a robust rostral-caudal (R-C) gradient of tactile information transmission in which rostral shorter vibrissae displayed a higher sensitivity and bigger differences in response to different textures, whereas longer caudal vibrissae were less sensitive. This gradient is evident in several dynamic properties of vibrissae trajectories. As rodents sample the environment with multiple vibrissae, we found that combining tactile signals from multiple vibrissae resulted in an increased sensitivity and bigger differences in response to the different textures. Nonetheless, we found that texture identity is not represented spatially across the whisker pad. Based on the responses of first-order sensory neurons, we found that they adhere to the tactile information conveyed by the vibrissae. That is, neurons innervating rostral vibrissae were better suited for texture discrimination, whereas neurons innervating caudal vibrissae were more suited for edge detection. These results suggest that the whisker array in rodents forms a sensory structure in which different facets of tactile information are transmitted through location-dependent gradient of vibrissae on the rat's face.

## Introduction

Facial whiskers, or vibrissae, are found in many mammalian species. They project outwards and forwards from the snout of the animal to form a tactile sensory array that surrounds the head [1]. This array is composed of 2 functionally distinct whisker systems—the long, moveable macrovibrissae, which are actively swept across objects and surfaces in a rhythmic forward and backward motion called whisking [2–4], and shorter, non-moveable microvibrissae [5–7]. In rodents, the macrovibrissae form a two-dimensional grid of 5 rows on each side of the snout, such that each row is composed of between 5 and 9 whiskers [5], whereas the shorter microvibrissae have a higher density and are more anterior.

**Funding:** This work was supported by a grant from the Israel Science Foundation 950/17 to RA (https://www.isf.org.il/). The funders had no role in study design, data collection and analysis, decision to publish, or preparation of the manuscript.

**Competing interests:** The authors have declared that no competing interests exist.

**Abbreviations:** ANOVA, analysis of variance; AP, anterior posterior; AUC, area under the ROC curve; BGU, Ben Gurion University; LFP, local field potential; ML, medial lateral; OWLA, Office of Laboratory Animal Welfare; PSTH, peri stimulus time histogram; R-C, rostral-caudal; RA, rapidly adapting; ROC, receiver operating characteristic; SA, slowly adapting; TG, trigeminal ganglion.

Rodents use their whiskers to detect and distinguish a variety of tactile features in their environment [8], including edge position [9, 10], shape [5, 11], aperture and gap width [12], and textures [7, 13–18]. Despite a long history of morphological, physiological, and behavioral work indicating that the whiskers are functionally distinct (Hartmann and colleagues [19, 20] and Moore and colleagues [21, 22]), many neurophysiological studies assume that the whiskers are functionally equivalent and that the array forms a diffuse spatial sensor. However, the evolutionarily conserved spatial arrangement of whisker arrays may have implications for probing the environment [5].

A few studies have addressed the functional architecture of the mystacial pad (i.e., the vibrissae pad). Several studies [5, 23] have suggested, based on behavioral data, that the mystacial macrovibrissae row is a "distance decoder." Its presumed function is to derive head-centered object shapes at the various angles represented by vibrissae rows while the microvibrissae are involved in object recognition tasks [7]. In recent years, it became more accepted that the vibrissae in different regions of the array are not interchangeable sensors but rather functionally grouped to acquire particular types of information about the environment [20]. Further support for this idea comes from several studies showing a gradient in macrovibrissae length, thickness, and whisking movement range. Thus, the caudal whiskers are longer and move at larger angles than rostral whiskers [5, 13, 24–26].

These whiskers' qualities express themselves as biomechanical characteristics, which have been utilized in several theories of tactile coding. "The resonance theory" [21, 27, 28] holds that each whisker's intrinsic dynamics (including but not limited to resonance) governs its texture-related response. According to this view, texture identity is represented spatially across the whisker pad. Whiskers' mechanical properties can also be utilized in alternative models in which textures are encoded by whisker patterns of movement that are induced by whiskers sweeping across surfaces. These patterns have been proposed to include both mean speed and spectral composition of whisker vibrations [29–31] and discrete high-acceleration micromotions called stick-slip events [32–36]. We posit here that whiskers' mechanical properties could impact all of these aspects, thereby differentially influencing the flow of tactile information across the pad.

Given the changes in whisker properties along the mystacial arcs, we sought to examine the functional implications of this gradient. Here, we describe the relative functional contributions of each individual whisker in the mystacial pad. We suggest that the mystacial macrovibrissae form a gradient of tactile information transmission in which longer caudal vibrissae are mainly involved in active edge localization, whereas the rostral shorter vibrissae transmit both edge collision and texture coarseness information. This structure has functional significance for tactile exploration and navigation.

## Materials and methods

### Animals and surgery

All experiments were conducted in accordance with international standards and were approved by Ben Gurion University (BGU) Committee for the Ethical Care and Use of Animals in Research. The project license is IL-71-11-2016. BGU's animal care and use program is supervised and fully assured by the Israeli Council for Animal Experimentation of the Ministry of Health. It is operated according to Israel's Animal Welfare Act 1994 and follows the Guide for Care and Use of Laboratory Animals (NRC 2011). In addition, BGU is assured by the Office of Laboratory Animal Welfare, USA (OWLA) (#A5060-01). Male Sprague Dawley rats (250–350 gm) were used. Surgical anesthesia was induced by urethane (1.5 gm/kg i.p.) and maintained at a constant level by monitoring forepaw withdrawal and corneal reflex. Extra

doses (10% of the original dose) were administrated as necessary. Atropine methyl nitrate (0.3 mg/kg i.m.) was administered after general anesthesia to prevent respiratory complications. Body temperature was maintained near 37˚C using a servo-controlled heating blanket (Harvard, Holliston, MA). After placing the subjects in a stereotactic apparatus (TSE, Bad Homburg, Germany), an opening was made in the skull overlying the trigeminal ganglion (TG), and tungsten microelectrodes (2 MΩ, NanoBio Sensors, Israel) were lowered according to known stereotaxic coordinates of the TG (1.5–3 medial lateral [ML], 0.5–2.5 anterior posterior [AP]) [37, 38] until units drivable by whisker stimulations were encountered. The recorded signals were amplified (×1,000), band-pass filtered (1 Hz–10 kHz), digitized (25 kHz), and stored for off-line spike sorting and analysis. The data were then separated into local field potentials (LFPs; 1–150 Hz) and isolated single-unit activity (0.5–10 kHz). All neurons could be driven by manual stimulation of one of the whiskers, and all had single-whisker receptive fields. Spike extraction and sorting implemented MClust (by A.D. Redish available from http://redishlab.neuroscience.umn.edu/MClust/MClust.html), which is a Matlab-based (Mathworks, Natick, MA) spike-sorting software. The extracted and sorted spikes were stored at a 1-ms resolution, and peri stimulus time histograms (PSTHs) were computed.

## Whisker stimulation

We replayed whisker movements across different surfaces by covering the face of a rotating cylinder with several grades of sandpaper with different degrees of coarseness and rotating the wheel against the whiskers (Fig 1A). The wheel face was placed so that the macrovibrissae rested upon it (Fig 1A). The wheel was placed to mimic rostral-caudal (R-C) whisker movement during head movement. The head velocities associated with rat exploration were taken from Lottem and Azouz [18].The velocity was controlled using a DC motor driven at approximately 39 mm/s to replicate median head velocity. The 30-mm-diameter wheel was driven by a DC motor (Farnell, Leeds, UK). We employed surfaces of different grades (from coarse-grained to fine-grained; the numbers in the parentheses indicate the average grain diameter): P120 (125 μm), P150 (100 μm), P220 (68 μm), P400 (35 μm), P600 (26 μm), P800 (22 μm), and P1200 (15 μm). These grades were chosen in accordance with previous studies [16, 29, 31]. For each texture, we recorded continuously for about 2 min per texture of the rotating cylinder. To examine TG neuronal responses to edges, we attach a 5-mm edge to the face of the wheel. We divided the data into segments in which each segment contained whisker colliding with the edge and texture. Each revolution of the wheel lasted for about 3 s, and no noticeable adaptation in neuronal firing rates were detected.

Whisker displacements transmitted to the receptors in the follicle were measured by an infrared photo-sensor (resolution: 1 μm; Panasonic: CNZ1120) placed about 2 mm from the pad. The voltage signals were digitized at 25 KHz and amplified (see [34] for principles of sensor operation and a description of sensor calibration). In some of recordings (where mentioned), whisker images were acquired with Mikrotron GmbH (Unterschleissheim, Germany) CoaXPress 4CXP camera at 1,600 frames/s; 4 megapixel resolution. The camera was installed above the arena and thus produced an overhead view. We rotated the texture-covered cylinders at velocities corresponding to head movements (see above). All the movies were analyzed using the Janelia whisker tracker software [39].

## Data analysis

To examine the influence of whisker identity on responses to edges and textures (Fig 1A and 1B), we separated edge and texture related stimulation epochs by identifying big excursion (mean ± 3 SD) in whisker vibrations during wheel rotation. In some experiments, we verified

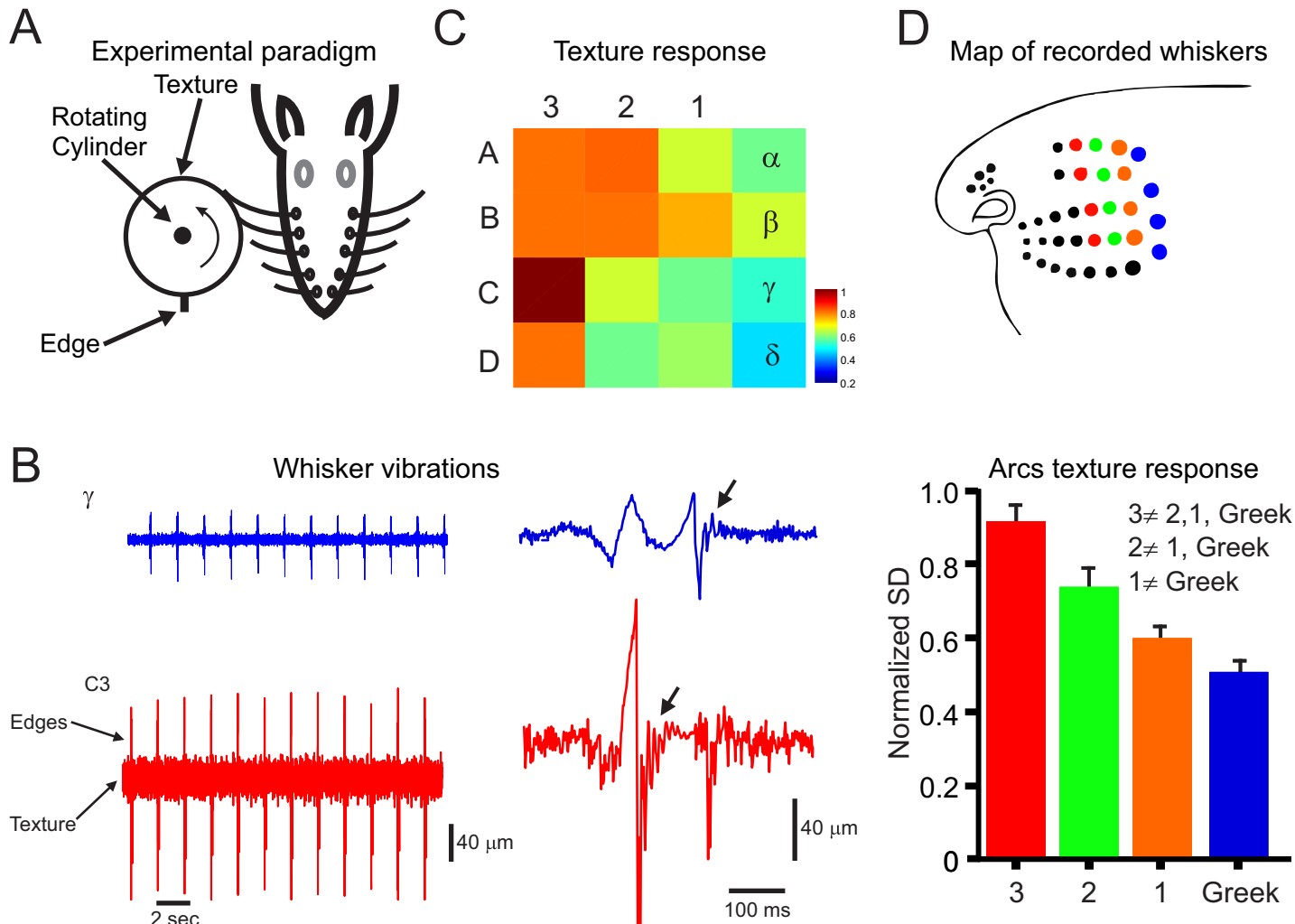

**Fig 1. Differential biomechanical characteristics of different whiskers arcs.** (A) Experimental design. The whiskers are contacting a rotating cylinder covered with textured sandpaper and an edge. (B) Example of 2 outermost whiskers' vibrations in response to texture (P220) and edges. On the right, a magnified segment from the left panel showing whiskers' motion while colliding with an edge and the resultant resonant movements (arrows). (C). Heat map of whiskers' response movement SD to all textures across all whiskers. Each pixel denotes the normalized whisker vibrations' SD to the highest value in the map. (D) The upper panel shows the recorded whiskers. The lower panel shows the mean and SD of each of the arcs in C. The inequality sign indicates a statistically significant difference between the various arcs. The underlying data for this Figure can be found in S1 Data.

edge collision by recording the interaction between whiskers and wheel with a high-frame rate video camera. A larger excursion of whisker movement was always associated with edges. Mean ± 3 SD was the criterion that could distinguish in all cases edges from surfaces. We then aligned the whisker responses and the corresponding neuronal responses to create PSTHs. Once PSTHs were created, we defined manually the temporal margins of edge responses. To determine the ratios between edge and texture responses, we calculated the corresponding firing rates.

To calculate whisker curvature that allows the calculation of forces acting on the whisker follicle [24, 40, 41], we used the method by Towal and colleagues [40] implemented in Janelia whisker tracker software [39]. For analysis, curvature was measured at 10 points along the whisker, and the maximum local curvature per image was extracted.

The significance of the differences between measured parameters was evaluated using one-way analysis of variance (ANOVA). When significant differences were indicated in the F ratio

test ($P < 0.05$), the Tukey method for multiple comparisons was used to determine those pairs of measured parameters that differed significantly from each other within a group of parameters ($P < 0.05$ or $P < 0.01$). The results are presented as mean ± SD. Error bars in all the figures indicate the SD unless otherwise noted. To avoid cluttering in some of the graphs, we use single-sided error bars.

### Receiver operating characteristics analysis

We used signal detection theory (receiver operating characteristics [ROC] analysis) [42] to compute the probability that an ideal observer could accurately determine the differences among the different textures based on neuronal activity. For each measured texture pair, an ROC curve was constructed. The ROC curve is a two-dimensional plot of hit probability on the ordinate against false-alarm probability on the abscissa. To transform raw data into a measure of discriminability, we analyzed the distributions of neuronal firing rates across trials. Green and Swets [42] showed that the area under the ROC curve (AUC) corresponds to the performance expected of an ideal observer in a two-alternative, forced-choice paradigm, such as the one used in the present analysis. The ROC curve was calculated for the firing rate of a single neuron as a function of texture. We then averaged all AUC values of all neurons, all texture pairs in the different rows or arcs.

The firing rate in trial $k$ is the spike count $n_k^{sp}$ in an interval of duration $T$ divided by $T$

$$Fr = \frac{n_k^{sp}}{T}$$

The length $T$ for texture signal was set to $T = 50$ ms and for edges $T = 10$ ms.

To measure the significance level of $P$(correct) in the ensemble of TG neurons, we took all possible textures comparisons for all neurons and shuffled the trials across the different stimuli. We then repeated this procedure 500 times. The significance level was set at 90% of this population, namely 0.53.

### Results

We examined the differential role of the macrovibrissae in the transmission of tactile information and quantitatively evaluated the mechanical and neuronal mechanisms underlying this process. We sought to examine whether the mystacial pad has a location-dependent differentiation of transmission of tactile information. We used several approaches to address this issue. First, we examined whether a gradient of whiskers' biomechanical characteristics across the pad impact their responses. Second, we studied whether this gradient is expressed in neuronal responses to textures. To evaluate whether the different whiskers' biomechanical properties play a role in tactile information transmission, we replayed passive whisker movements across different surfaces by covering the face of a rotating cylinder with several grades of sandpaper with different degrees of coarseness. The cylinder face was placed orthogonally to the vibrissae so that the vibrissa rested upon it (Fig 1A). These surfaces were placed at 2 different distances from the pad (whisker tip and 5 mm closer to the pad) and at 2 different wheel velocities (39 mm/s and 50 mm/s).

We examined the influence of the biomechanical properties of the vibrissae on the transformation of surface coarseness into whisker vibrations by measuring across several anaesthetized animals ($n = 5$; from both sides in some animals) for each vibrissa. In Fig 1B, we show an example of whisker vibrations in 2 vibrissae in response to P220 texture and an embedded edge. These panels show that rostral shorter, slender vibrissae expressed increased range of vibrations for textures and edges, whereas the caudal vibrissae presented lesser variance. To

establish a functional map of the mechanical properties of the vibrissae, we first calculated whiskers' position SD of each of the measured whisker vibrations in response to all studied textures. The whisker vibrations' SD measure reflects the magnitude of stick-slip events when whiskers scan across a surface. We then normalized each texture SD to the maximal value across all whiskers and averaged the normalized values for each whisker in the map. That is, each pixel value in the maps in Fig 1C is an average of normalized SD for all textures across all whiskers. In Fig 1 and S1 and S2 Figs, vibrissae movements were measured in response to P220 sandpapers. A relatively smaller variance indicates a larger attenuation of the texture signal. This analysis revealed a location-dependent gradient in which whisker vibrations were transmitted more robustly through rostral whiskers (Fig 1C). Each pixel in this plot represents an average of numerous vibrissae [Greek:(α: $n = 5$; β: $n = 5$; γ: $n = 3$; δ: $n = 5$); arc 1: (A: $n = 5$; B: $n = 5$; C: $n = 5$; D: $n = 5$); arc 2: (A: $n = 5$; B: $n = 5$; C: $n = 5$; D: $n = 5$); arc 3: (A: $n = 5$; B: $n = 5$; C: $n = 5$; D: $n = 5$); due to size limitation of the sensor and the short length of rostral whiskers, we did not record whiskers' displacements and corresponding neuronal responses to whiskers rostral to arc 3]. We then averaged all values in each arc across all animals and examined the statistical validity of the R-C gradient (Fig 1D upper panel). We found statistically significant differences between most arcs (Fig 1D lower panel). These results suggest a caudal-rostral functional gradient that arises from the biomechanical differences of the sensing organs (i.e., the whiskers). To examine the robustness of this gradient, we changed wheel velocity and the proximity of the wheel to the pad (S1 Fig) and found minor changes in the gradient. Moreover, we examined whether this gradient exists in the dorsal-ventral plane (S2 Fig) and found no significant gradient in this plane.

To examine whether the gradient in whiskers' mechanical properties is instrumental in texture discrimination, we compared whisker vibrations to 5 textures (P120, P220, P600, P800, and P1200) in the different arcs. We examined 3 different measures of whisker vibrations in response to these textures: whisker position SD, spectral composition of whisker vibrations, and whisker curvature SD, which allows the calculation of forces acting on the whisker follicle [24, 40, 41]. Examples of these characteristics in C4 and γ whiskers in response to P120, P600, P1200 textures are shown in Fig 2A, 2C and 2E. These examples show that, in all measured response characteristics, the range of whisker responses is higher in the C4 whisker compared with the γ whisker. Verification of this notion is shown in the average for C4, C2, and γ whiskers ($n = 5$) in Fig 2B, 2D and 2F. We quantified the range of whisker vibration in response to the different textures by calculating the 3 response characteristics. The figures show a gradual relation between texture coarseness and whisker response characteristics across the whisker pad, i.e., coarser and finer surfaces are expressed in a higher and lower response properties value, respectively. Moreover, rostral whiskers respond with a bigger range of whisker responses than caudal whiskers.

To quantify this differential sensitivity across the pad, we used 2 approaches: first, we plotted the relations between mean particle diameter (P120: 125; P220: 68; P600: 25.8; P800: 21.8; and P1200: 15.3; mean particle diameter in μm) and the 3 whiskers' response parameters (each point in these plots is normalized to the maximal value). To quantify the difference between textures in the different whiskers, we calculated the linear regression fit for each of the response characteristics for each whisker as a function of mean particle diameter in each texture (Fig 3A, 3C and 3E). We found that the slope was steeper in C4 than in γ whiskers in all measures. Fig 3B, 3D and 3F show this in all recorded whiskers. The normalized slopes (normalized to C4 slope) show that rostral whiskers display a bigger difference between textures than caudal whiskers. Second, we employed ideal observer analysis to quantify the discriminative power of each of the whiskers. We divided the continuous whisker signals into multiple segments (trials; 500 ms in duration) and used the AUC measure to calculate—for each

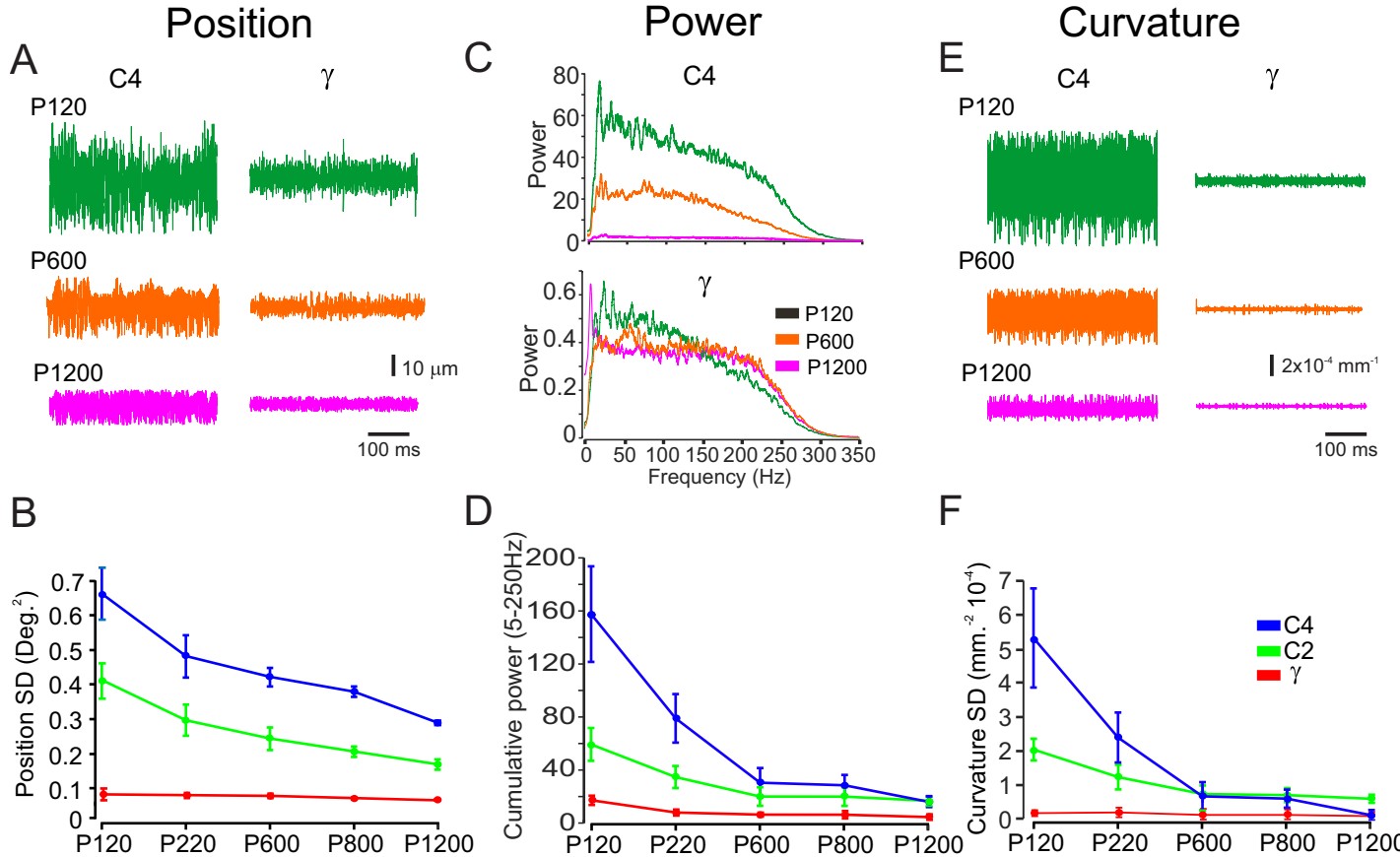

**Fig 2. Rostral whiskers exhibit higher responses range to different texture grain.** (A). Example of 2 outermost whiskers' (C4, γ) vibrations in response to textures (P120, P600, P1200). (B) The panel shows the quantification of 3 whiskers' vibrations' SD in response to different textures. (C) Power spectra of movements of the same whiskers across different textures is shown color coded. (D) The panel shows the quantification of 3 whiskers' vibrations' cumulative power in the 5–250 Hz range in response to different textures. (D) The intrinsic curvature of 2 outermost whiskers in response to the same textures as in A. (E) The panel shows the quantification of 3 whiskers' curvature SD in response to different textures. These examples show a higher sensitivity to the different textures in rostral whiskers in all of these parameters. The underlying data for this Figure can be found in S1 Data.

whisker—the average AUC across all texture combinations. We found that rostral whiskers, using all measures, could better discriminate between the different textures (Fig 4A–4C).

Since rodents sample the environment with multiple whiskers, we examined whether combining tactile signals from multiple whiskers improves texture discrimination capabilities of the system. Specifically, we studied whether integrating texture information from multiple whiskers will increase texture discrimination as measured by the AUC for ROC curve. An example of such an ROC curve is shown in S3 Fig for γ, C2, and γ + C2 whiskers. Initially, we averaged the responses to textures from multiple whiskers and then calculated the AUC. Fig 4D and 4E show the results of these calculations for γ and C4 whiskers with addition of information from more rostral and caudal whiskers, respectively. We found that adding information to γ whisker from rostral whiskers improved texture discrimination. Integration of up to 2 whiskers away (γ + C1; γ + C2; γ + C1 + C2) resulted in bigger AUC than each of the more rostral whiskers. Beyond 2 whiskers, the AUC was smaller than each individual whisker separately (compare γ + C3 and γ + C4 in Fig 4D to arc 3 and arc 4 in Fig 4A). The same applies for the C4 whisker: integration of up to 2 whiskers away (C4 + C3; C4 + C2; C4 + C3 + C2) did not result in smaller AUC of the C4 whisker. Beyond 2 whiskers, the AUC was smaller than

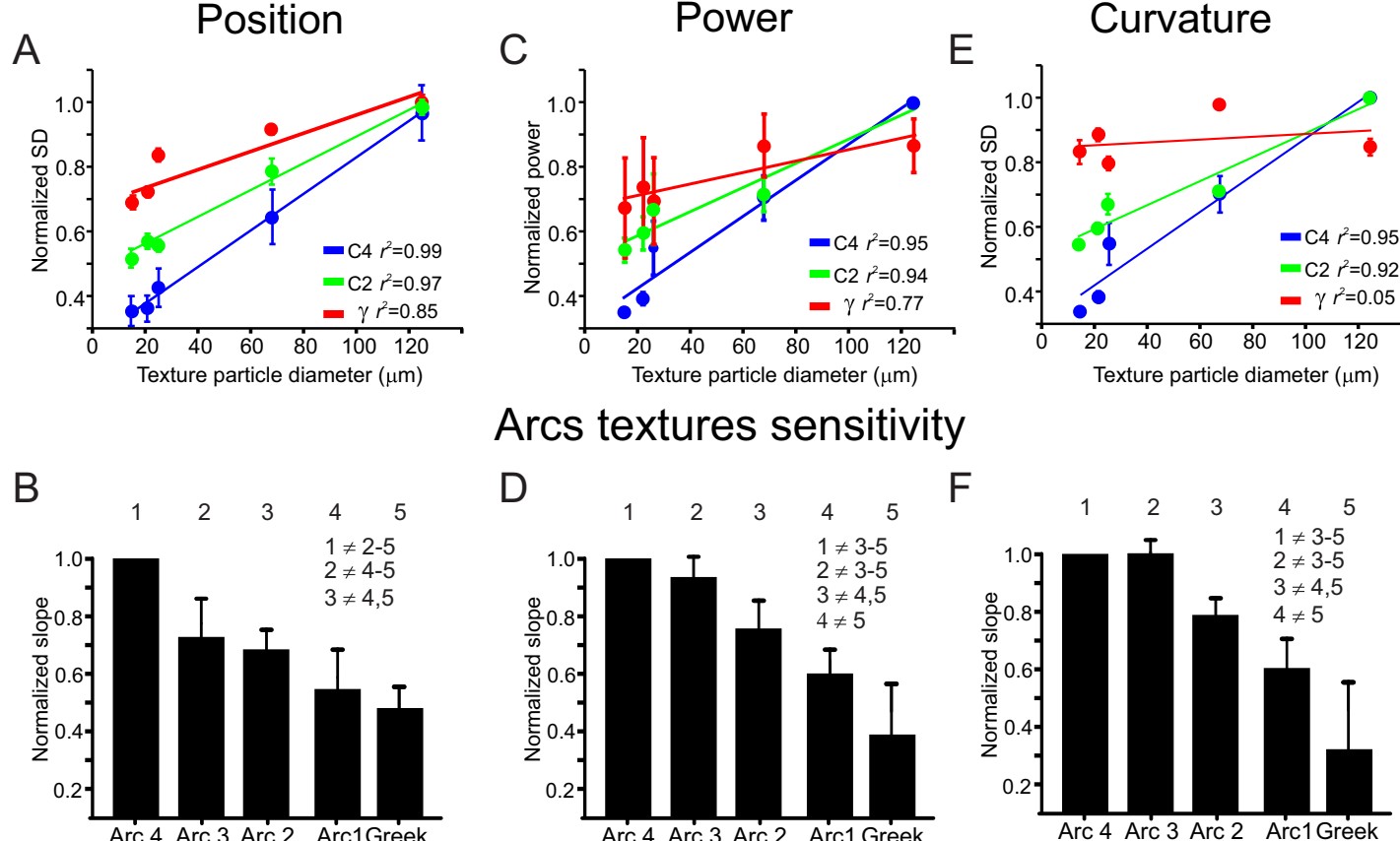

**Fig 3. Rostral whiskers exhibit higher response sensitivity to texture grain.** (A, C, E) Linear regression fit of the normalized SD (A), power (C), and curvature (E) of whisker vibrations in response to several textures ($n = 5$; each texture is represented by the mean particle size). (B, D, F) Normalized slopes (normalized to the slope of arc 4 whiskers of each session) of the linear regression fit for all whiskers. The numbers indicate the different arcs. The inequality sign indicates a statistically significant difference between the various arcs. The underlying data for this Figure can be found in S1 Data.

AUC of the C4 whisker (compare C4 + C1, C4 + γ in Fig 4E to arc 4 in Fig 4A). To determine whether this integration occurs linearly, we devised a linearity measure in which we calculated the ratio between the AUC of the average (Fig 4D and 4E, see above) and the average AUC (Fig 4A). We found that in most cases, the AUC sums supralinearly (Fig 4F). Together, we conclude that the mechanical properties of the pad whiskers enable them to transmit differential aspects of tactile information. Moreover, integrating tactile signals from multiple whiskers might improve the capabilities of the whisker somatosensory system in texture discrimination.

When rats are whisking against a textured surface, owing to whiskers having different lengths, the encounter with an object may occur at different locations along the shaft for different whiskers. Moreover, caudal longer whiskers also sweep across a wider range of motion, which could lead to an increase in the forces exerted on objects during active whisking. The increased forces applied by the longer whiskers during these 2 situations may result in a diminution of the mechanical map that was described earlier. To address this issue, we changed the proximity of the wheel to the pad to keep the distance from the pad constant. Fig 5A depicts the experimental paradigm and whiskers C3 and γ vibrations in response to P220 at the same distance from the pad. We then measured the whiskers' response to this texture by monitoring whiskers' position as well as curvature. We then quantified the SD of whisker position and curvature SD in this animal and found a gradual decrease of whiskers' responses as a function of

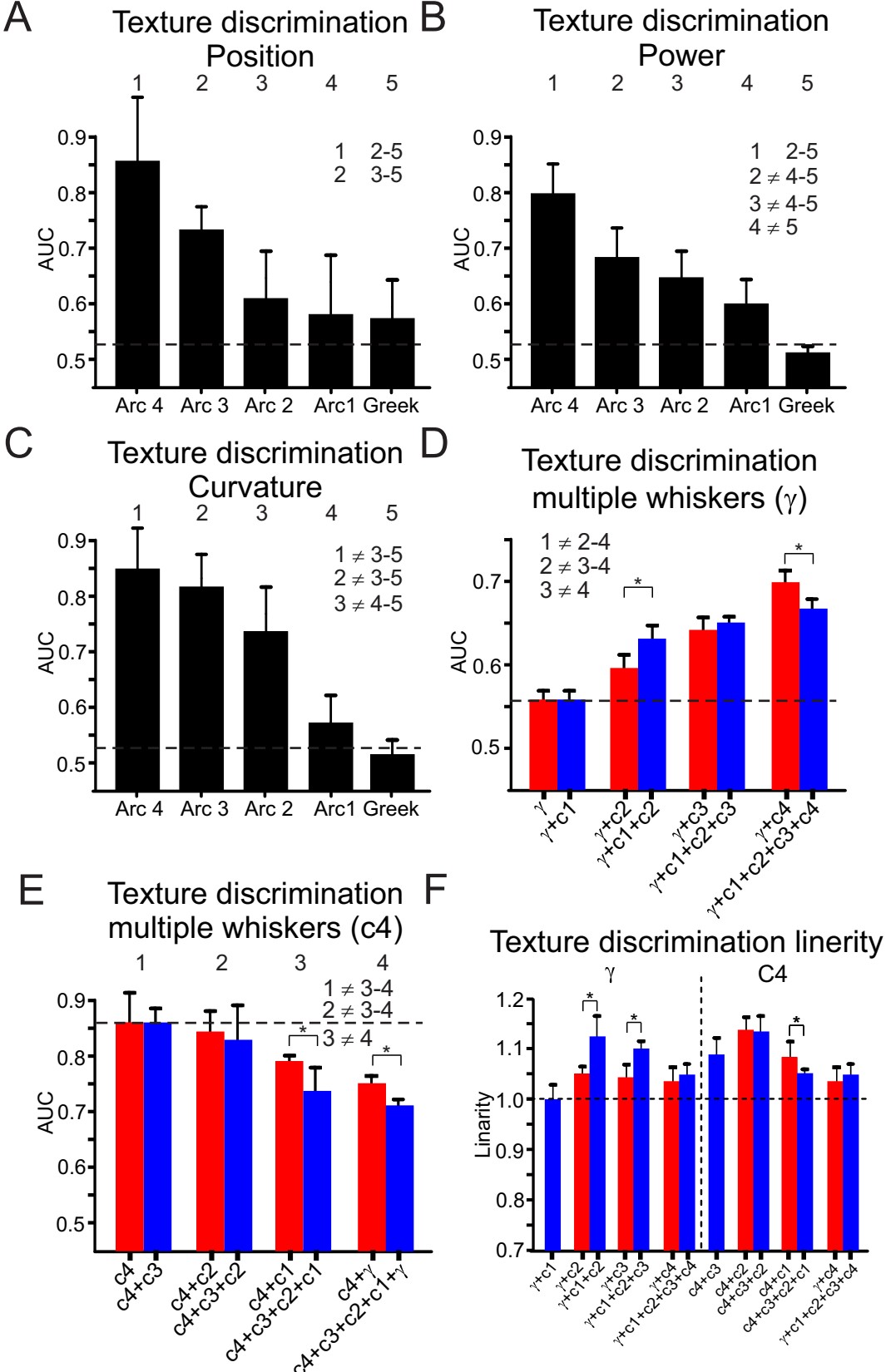

**Fig 4. Rostral whiskers exhibit greater texture discrimination.** Average AUC for texture discrimination as a function of the different arcs. Whisker position SD (panel A), power (panel B), and curvature (panel C). (D–E) Combining tactile signal from multiple whiskers increased texture discrimination for γ whisker but not for C4. (F) A measure of linearity of summation of signals from multiple whiskers (see text for details). In most cases, summation in supralinear. The numbers indicate the different arcs. The inequality sign indicates a statistically significant difference between the various arcs. The underlying data for this Figure can be found in S1 Data. AUC, area under the receiver operating characteristic curve.

whisker identity (Fig 5B and 5D). Averaging across all animals (*n* = 6), we found a consistent decrease in whiskers' responses for more caudal whiskers (Fig 5C and 5E). These results indicate that the gradient of tactile properties across the pad is robust and maintained during different conditions.

We sought to test how the gradient of biomechanical properties across the pad directly translates into neuronal activity in first-order sensory neurons by examining the relations between neuronal responses to textures and edges in the different whiskers. We used several approaches to examine this gradient. First, we recorded responses to step whisker deflections from 71 TG neurons obtained from 16 adult rats. As previously described [38, 43–46], such neurons respond to step stimuli with either a phasic response, firing only to stimulus onset/offset, or a phasic-tonic response, firing both at stimulus onset/offset and throughout the stimulus hold period. These firing patterns are conventionally used to characterize neurons as either rapidly adapting (RA) or slowly adapting (SA), respectively. We did not find any significant bias in neuronal types towards any of the arcs (Table 1). Second, we recorded extracellularly from 57 TG neurons (Greek: *n* = 13; arc 1: *n* = 15; arc 2: *n* = 9; arc 3: *n* = 10; arc 4: *n* = 11). The experimental paradigm is shown in Fig 1A. An example of neuronal responses of 4 separate TG neurons to a rotating wheel covered with textures and an edge is shown in the upper panels in Fig 6A. We initially divided the continuous neuronal recordings (Fig 1B) into segments that contained an epoch preceding response to edges, responses to edge, and an epoch containing responses to textures (Fig 6A, red boxes). Demarcation of these epochs within each segment was done using the criteria from whisker signals (see Materials and Methods section). Once these segments were created, we created raster plots and their corresponding PSTHs for each neuron for the different conditions (Fig 6A). The PSTHs of these neurons indicate that the caudal whiskers' neurons responded mainly to edges, whereas progressively more rostral whiskers' neurons responded to both edges and texture stimuli. This was further verified by calculating the normalized ratio between discharge probability in response to edges and textures (normalization was done as the ratio between each whiskers value and the maximal value, which was always the Greek arcs). Fig 6B left panel shows that this ratio decreased gradually from caudal to rostral, since texture-associated firing increased gradually when moving to whiskers that are more rostral. To examine the robustness of this phenomenon, we repeated these experiments at a closer distance from the pad (Fig 6A, lower panels). Decreasing the distance from the pad resulted in an increase in neuronal firing probability. Nonetheless, the ratio decreased gradually from caudal to rostral whiskers (Fig 6B, middle and right panels). Finally, we compared the firing rates across the different rows and didn't find any statistically significant differences in their firing rates (see Materials and Methods section for statistical significance of the differences between measured parameters; Fig 6C). Our results indicate that, irrespective of surface distance, the ratio between discharge rates in response to edges and textures does not change qualitatively.

To examine the influence of whiskers' biomechanical properties on tactile information transmission, and whether these characteristics impact neuronal ability to discriminate between different textures as well as between textures and edges, we initially divided the data into 2 epochs: neuronal responses to fine-grained textures and edges (Fig 6A). We then

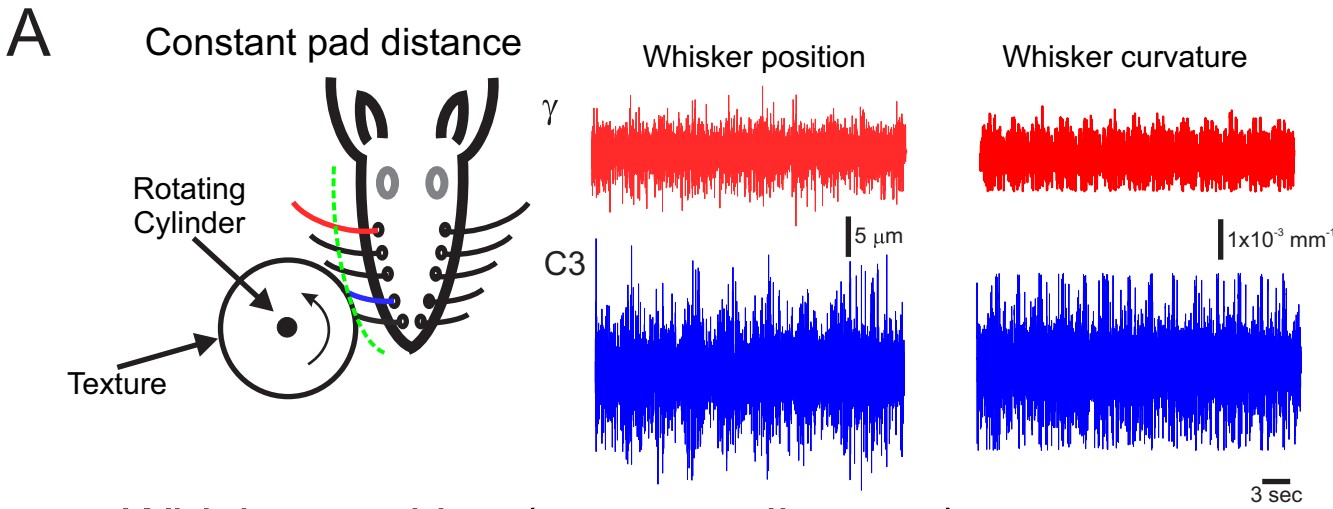

## Whisker position (constant distance)

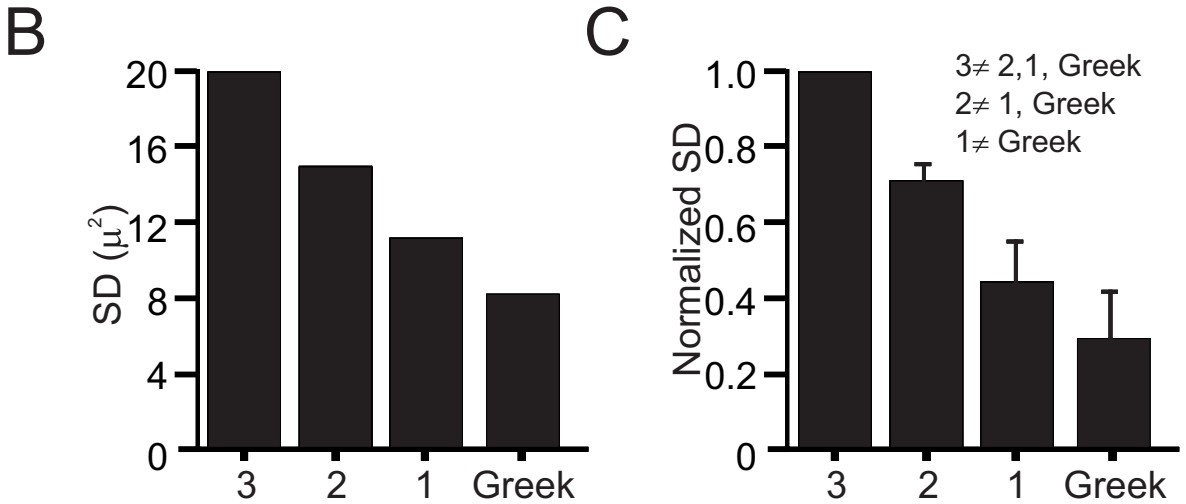

## Whisker curvature (constant distance)

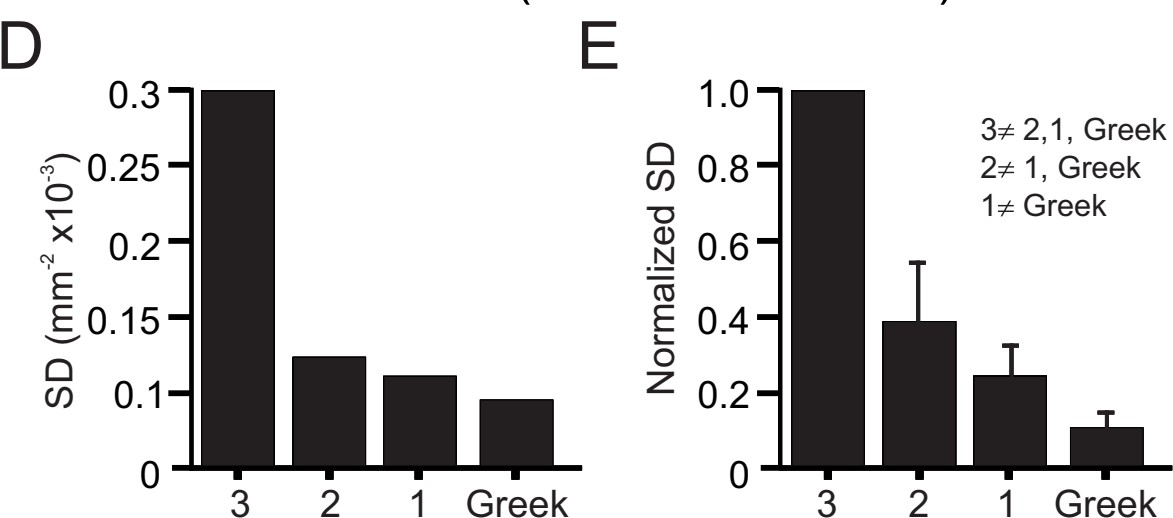

**Fig 5. Changes in surface distance does not alter the gradient of tactile inputs across the pad.** (A) Experimental design. The whiskers are contacting a rotating cylinder covered with textured sandpaper at the same distance. Example of 2 outermost whiskers' (C3 and γ) vibrations and curvature in response to texture (P220). (B–C). The mean SD of each of the arcs in panel A. Normalized SD mean and SD of each arc in all animals ($n = 5$). (D–E) The same as in panels B–C for curvature. The inequality sign indicates a statistically significant difference between the various arcs. The underlying data for this Figure can be found in S1 Data.

calculated the firing rates for each texture and edge epoch. An average firing rate (for each texture epoch in the neuronal response) of 2 neurons taken from the same animal for β and C3 whiskers is shown in Fig 7A; the panel shows a steeper change in firing rates across textures in C3 neurons, whereas β neurons only slightly changed their firing rates with the different textures. We then calculated the normalized firing rates (normalized to highest firing rates across all whiskers) and found a steeper slope for firing rates in arc 3 versus Greek ($n = 23$). To quantify the relations between surface coarseness and firing rates in all neurons across all arcs, we calculated the slope of the linear regression between average particle diameter (see Fig 3A; P150 = 100 μm; P220 = 68 μm; P400 = 35 μm; P800 = 22 μm) and normalized firing rates for each neuron. We then normalized the slope of each neuron to the steepest slope in the sample ($n = 57$) and divided the values to the corresponding arcs. We found a steeper slope for the rostral whisker then the caudal whisker, suggesting that neurons innervating rostral whiskers are better suited for discriminating between textures. To examine this premise, we used the ideal observer approach (see Materials and Methods) and compared the mean AUCs across all textures' pairs for all whiskers ($n = 57$). We found that TG neurons in rostral whiskers have significantly higher AUCs, suggesting that they perform better in fine-grained texture discrimination (Fig 7D, blue bars). In contrast, comparing the firing rates of fine-grained textures and edges revealed that TG neurons in caudal whiskers have significantly higher AUCs for texture-edge discrimination (Fig 7D, red bars). To examine the robustness of this phenomenon, we repeated these measurements in a different set of whiskers at a closer distance of the wheel to the pad (S4D Fig). Together, these results indicate a differential role for caudal and rostral whiskers in texture and edge discrimination. Specifically, the longer, more movable caudal whiskers are more suitable for distinguishing between collision with edges and vibrations associated with texture, and the shorter, less movable rostral whiskers are more suitable for discriminating fine surface details.

## Discussion

Rodents are strongly tactile animals that scan the environment with their macrovibrissae and orient to explore edges further with their shorter and more densely packed microvibrissae [5, 7, 26]. The facial vibrissae are organized into a pattern of several arcs and rows [5, 47]. We posit that this pattern could serve a functional role in the way that animals sense the environment. Here, we focused on the function of this orderly array of facial whiskers and addressed the issue of the relations between the structural features and functional operations of the vibrissae apparatus. We suggest that the mystacial whiskers form a gradient of tactile

**Table 1. Distribution of neuronal types across all whiskers.**

| Neuronal type \ Arc | Greek ($n = 20$) | 1 ($n = 15$) | 2 ($n = 16$) | 3 ($n = 15$) | 4 ($n = 5$) |
|---|---|---|---|---|---|
| RA | 9 | 8 | 9 | 7 | 3 |
| SA | 11 | 7 | 7 | 8 | 2 |

**Abbreviations:** RA, rapidly adapting; SA, slowly adapting

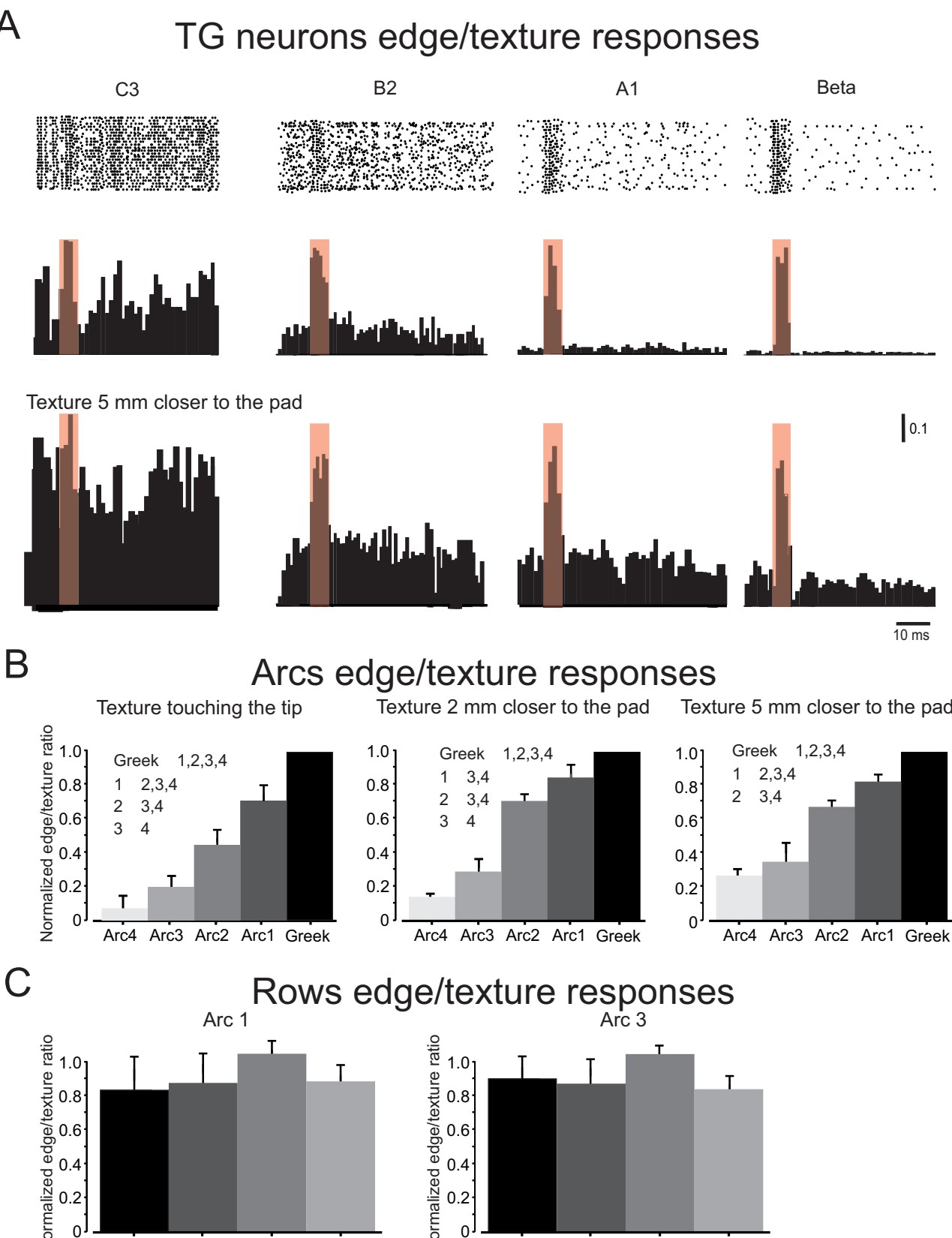

**Fig 6. TG neurons innervating rostral whiskers exhibit higher response range to different texture grain.** (A) Typical examples of raster plots and PSTHs from different arcs aligned to edge contact shown from caudal (right) to rostral (left). The red transparent square demarcates edge-related responses. The middle and lower panels show the responses of the neurons at different distances of texture wheel from the pad. (B) Normalized firing rate ratio (the ratio between firing rates in response to edges and responses to textures normalized to the highest value across arcs) from the different arcs and at different conditions. (C) Normalized firing rate ratio (the ratio between firing rates in response to edges and responses to textures normalized to the highest value across rows) from 2 different rows. Scale bar of the PSTHs refers to firing probability at a given bin. The underlying data for this Figure can be found in S1 Data. PSTH, peri stimulus time histogram; TG, trigeminal ganglion.

information transmission. In this scheme, longer caudal macrovibrissae transmit mainly "where" information, whereas more rostral shorter vibrissae transmit both "where" and "what" information. Our results suggest that whisker array in rodents forms a sensory structure in which different aspects of tactile information are transmitted through a location-dependent gradient of vibrissae on the rat's face. One major feature that stems from our results is that, within a single whisk or object touch, rats may sample a region of an object at multiple spatiotemporal scales simultaneously. This may enable rodents to extract complex object features in a single whisk. The functional architecture of the pad ensures that vibrissae in different columns sample objects at different resolutions. Specifically, the rostral vibrissae sample objects with a higher spatiotemporal resolution than caudal vibrissae; the caudal vibrissae sample mostly edges. This is consistent with the possibility that the rostral whiskers immediately surrounding the snout may serve as a high-acuity "fovea" during tactual exploratory behavior [2, 7, 48]. This hypothesis supports the notion that whisking strategies are dependent on the behavioral task as well as the structure of the pad. The morphology of the vibrissae array has critical implication on the nature of the neural computations that can be associated with extraction of edge location and features. Thus, the differential whiskers' biomechanical properties in which length, thickness, and shape [49] are critical suggests the possibility for a parallel transduction of edges' location, shape, and texture through independent sensory channels [50].

## Mystacial pad anatomy

The anatomical basis for the gradient in whisking may depend on the organization of the muscles in the mystacial pad in relation to the whisker follicles and surrounding tissue. Movement of the whiskers is controlled by the facial motor nerve, which innervates 2 classes of muscles: the intrinsic and extrinsic muscles. In awake animals, whisker retraction probably involves the activation of extrinsic facial muscles, while protraction involves the intrinsic muscles [2]. The differential whisking amplitude across the pad could result from the anatomical structure of pad muscles, that is, the superficial extrinsic facial muscles could pull more caudal whiskers further back during retraction, having less of an impact on the more rostral whiskers. This will result in a larger movement of the more caudal whiskers. Alternatively, Haidarliu and colleagues [51] studied the nerve subdivision and the numerous mystacial pad muscles in the rat extensively and found a gradient in which the caudal whiskers are surrounded by larger intrinsic muscles and a larger subdivision of the extrinsic muscles. All these make the caudal vibrissae better suited for active edge localization. In addition to this global whisking control, several studies have shown that rats have sufficient degrees of freedom of whisker control to effect some differential movement of either individual whiskers or whisker columns [2, 52–54]. This individual follicle control may be the basis for complex kinematics and rich variability in whisking behavior [13, 26, 54–64].

## Texture discrimination

We found that whiskers, which are the first stage of tactile information translation, form a location-dependent gradient in which rostral whiskers show smaller attenuation in the

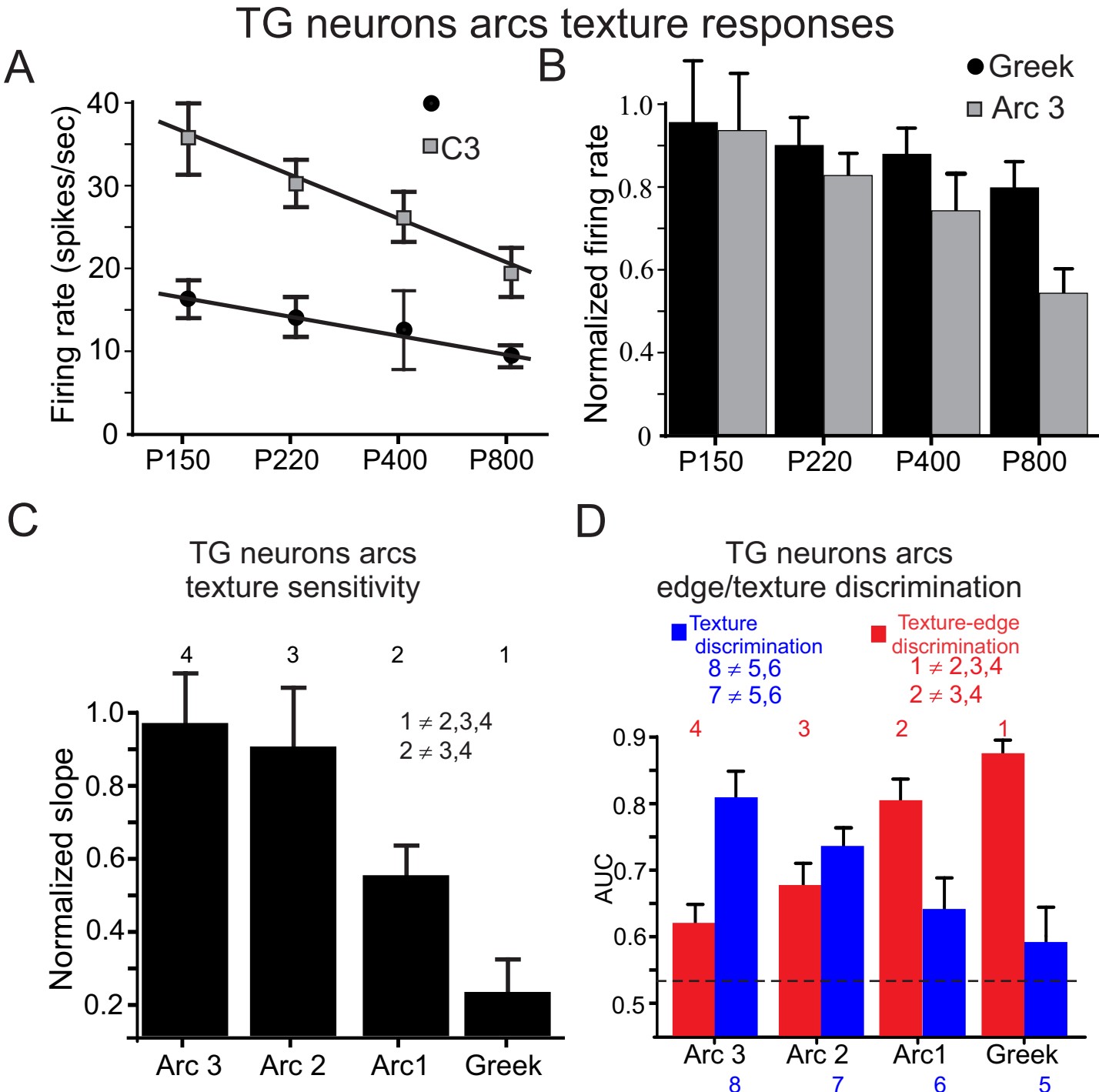

**Fig 7. TG neurons innervating rostral whiskers exhibit higher response sensitivity to texture grain.** (A) The relationship between texture coarseness and TG firing rates in 2 arcs. (B) Normalized firing rates (normalized to the maximal firing rates) for all neurons innervating Greek and arc 3 whiskers in response to the different textures. (C) Normalized slopes of the linear regression fit for normalized firing rates versus texture surface coarseness. (D) AUC for texture discrimination (blue bars) and texture-edge discrimination (red bars) for the different arcs. The numbers indicate the different arcs. The inequality sign indicates a statistically significant difference between the various arcs. The underlying data for this Figure can be found in S1 Data. AUC, area under the receiver operating characteristic curve; TG, trigeminal ganglion.

transformation from texture coarseness to whisker vibrations, whereas caudal whiskers mainly transmit larger signals such as edge collision (Figs 1–5). Moreover, we found that this map, which creates a gradient across the pad, is composed of multiple facets of whiskers' trajectories in response to textures and edges. The first is whisker position variance, in which we have shown that the variance in whiskers' position is a good predictor of surface coarseness, and the R-C gradient of the differential sensitivities suggests a plausible mechanism for texture coding. We posit that that this variance may reflect the magnitude of stick-slip events, which are related to neuronal discharge probability [65]. The second is the spectral composition of whisker vibrations or the modulation power—i.e., the power of whisker movement representing the modulation due to the texture surface [31]. Finally, whiskers' curvature plays an important role in shaping information transmission along the whisker to mechanoreceptors in the follicle by affecting the forces and moments at the vibrissal base [40]. We therefore used the variance in whisker curvature as a proxy for changes in bending moment [66]. Whisker-pole contacts caused substantial whisker bending, partially correlated with the whisker angle [67] and robust spiking. Thus, all of these response characteristics show this R-C gradient, suggesting multifaceted, multichannel streams of tactile information serving the animal in discrimination and detection tasks. These findings are supported in a recent study by Zuo and Diamond [68], which measured several dynamic properties of the whiskers' trajectories in awake behaving rats. Some of these features, averaged over the duration of a touch, covaried well with texture. These whisker dynamic features predicted both actual texture and the animal's choice. In addition to these multifaceted, multichannel streams of tactile information, we found that combining information from multiple whiskers improved the performance of the system in texture discrimination ([31] Fig 4D–4F). This occurred up to 2 whiskers away, alluding to an advantage for local integration. This has been shown in the barrel cortex, in which the dynamics of whisker deflections can greatly change when whiskers touch different textures or objects in the environment, resulting in global statistics at the scale of the whisker pad that may alter the functional properties of individual neurons [69]. The robustness of this gradient irrespective of distance, whisker velocity (S1 Fig; S2 Fig), and the torque exerted on the follicle (Fig 5) suggests a conserved mechanism in conveying tactile information to the neuronal system and might allude to multiple uses in which perceptual constancy is one.

We found an AP gradient of tactile information transmission in which TG neurons of longer caudal vibrissae mainly transmit edge collision information, whereas TG neurons of rostral shorter vibrissae transmit both edge collision and texture coarseness information (Figs 6 and 7). Our results suggest that these differential neuronal responses may stem from the mechanical properties of the whiskers and not from a change the neuronal characteristics of the mechanoreceptors of the different whiskers. Moreover, based on neuronal firing rates, we found that rostral whiskers' neurons are better suited for finer details of the tactile environment, whereas caudal whiskers' neurons are better suited for coarser details such as edge collision (Figs 6 and 7). Several important implications arise from these results. Since all tactile information available to the whisker somatosensory system originates in these mechanoreceptors, it is conceivable that the gradient of tactile information transmission on the vibrissa pad is expressed as a somatotopic cortical map that describes multiple spatiotemporal scales extending along the representations of arcs of vibrissae, much like the resonant theory [22]. Moreover, we posit that being exposed continuously to these diverse inputs may shape differentially both the wiring and the properties of the constituent neurons all along the pathway up to the cortex [70].

While we found a gradient of mechanical properties along the whisker pad which were reflected as texture-related response, our results do not show that texture identity is represented spatially across the whisker pad. That is, we could not support the hypothesis that only a specific set of textures will cause a group of whiskers to vibrate at their distinct natural

frequency, making a set of whiskers selective for these particular textures, thus splitting the tactile vibration signals into labeled frequency lines in the cortex.

## Methodological considerations

In the current study, whiskers were stimulated passively, thus our results may be accounted for by tactile inputs caused by head movements [18]. Nevertheless, the carriers of the tactile signal (stick-slip events) were preserved in both conditions [18, 33]. More importantly, the present work addressed a rather simple configuration in which the whiskers were passively dragged along a surface near-parallel to the mystacial pad, while exploration generally involves active whisking onto an edge located in front of the animal's snout. Such configurations involve complex three-dimensional whisker dynamics [60]. Moreover, while we used several characteristics of whisker vibrations to define the gradient of biomechanical properties across the pad and consequently the map of tactile properties, we are aware that whiskers with varying lengths or diameters will have different vibrational characteristics that are not captured in this one-dimensional characterization (see, for example, [50]). However, the findings of the map in the discharge rates of TG neurons suggest that the vibrational characteristics that were used in the current study are sufficient.

## Supporting information

**S1 Fig. Robustness of whiskers' properties map.** (A) To examine to robustness of the map in Fig 1, we changed the distance of the wheel to the pad (texture 5 mm closer to the pad; upper left panel), and wheel velocity (velocity approximately 50 mm/s; upper right panel). The lower panels show the mean and SD of each arc in the upper panels. The inequality sign indicates a statistically significant differences between the various arcs. The underlying data for this Figure can be found in S1 Data.
(DOCX)

**S2 Fig. Mechanical characteristics of different whiskers' rows are not different from each other.** (A) The panel shows the recorded rows. (B) The panels show the mean and SD of each of rows in panel A at different distances from the pad and at different wheel velocity. The biomechanical properties of the whiskers across the different rows and in the different conditions are not significantly different from each other. The underlying data for this Figure can be found in S1 Data.
(DOCX)

**S3 Fig. ROC curve for discrimination between P120 and P220 textures for γ (blue), C2 (red), and their combination.**
(DOCX)

**S4 Fig. A robust gradient of textures and edge-texture discrimination capabilities in TG neurons.** (A) Similar to Fig 7A, but textures are 5 mm closer to the pad. (B) AUC for texture discrimination (blue bars) and texture-edge discrimination (red bars) for the different arcs. (C) Similar to panel A but for rows. (D) Similar to panel B but for rows. The numbers indicate the different arcs. The inequality sign indicates a statistically significant difference between the various arcs. The underlying data for this Figure can be found in S1 Data.
(DOCX)

**S1 Data. Data from all main and supporting figures.**
(XLSX)

## Author Contributions

**Conceptualization:** Rony Azouz.

**Data curation:** Erez Gugig, Hariom Sharma, Rony Azouz.

**Formal analysis:** Erez Gugig, Hariom Sharma, Rony Azouz.

**Funding acquisition:** Rony Azouz.

**Investigation:** Rony Azouz.

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
