## [Editor Report · Decision Letter 0]

24 Feb 2020

Dear Rony, 

Thank you for submitting your manuscript entitled "Gradient of Tactile Properties in the Rat Whisker Pad" for consideration as a Research Article by PLOS Biology.

Your manuscript has now been evaluated by the PLOS Biology editorial staff, as well as by an academic editor with relevant expertise, and I'm writing to let you know that we would like to send your submission out for external peer review.

Please re-submit your manuscript within two working days, i.e. by Feb 26 2020 11:59PM.

Kind regards,

Roli

Senior Editor

PLOS Biology

---

## [Decision Letter · Decision Letter 1]

18 Apr 2020

Dear Rony,

Thank you very much for submitting your manuscript "Gradient of Tactile Properties in the Rat Whisker Pad" for consideration as a Research Article at PLOS Biology. Your manuscript has been evaluated by the PLOS Biology editors, an Academic Editor with relevant expertise, and by three independent reviewers.

You'll see that all three reviewers are broadly positive about your study, but each has a significant number of requests for additional work. For example, two reviewers question the utility of Fig 8, and reviewer #1 would like to see more raw data presented, and queries some statistical choices. Reviewers #2 and #3 ask for further analyses; the Academic Editor thought that reviewer 2's idea for studying multiple whiskers simultaneously seemed reasonable, and could give further interesting results, and reviewer #3 asks that you incorporate calculations of whisker bending forces (using published data).

In light of the reviews (below), we will not be able to accept the current version of the manuscript, but we would welcome re-submission of a much-revised version that takes into account all of the reviewers' comments. We cannot make any decision about publication until we have seen the revised manuscript and your response to the reviewers' comments. Your revised manuscript is also likely to be sent for further evaluation by the reviewers.

We expect to receive your revised manuscript within 2 months. 

**IMPORTANT - SUBMITTING YOUR REVISION**

*Re-submission Checklist*

*Published Peer Review*

*PLOS Data Policy*

*Blot and Gel Data Policy*

Best wishes,

Roli

Senior Editor

PLOS Biology

REVIEWERS' COMMENTS:

Reviewer #1:

As nocturnal animals, rodents rely on their array of whiskers to navigate and to collect information about the objects around them. The whiskers are organized on the snout in an orderly manner with the longest whiskers positioned most caudally and the whisker diameter and length decreasing systematically in the rostral direction. Here, Gugig, Sharma and Azuz study the implications of this gradient of biomechanical characteristics across the whisker pad, on the ability of the whisker to transmit texture/edge information, and quantify the response of first-order neurons to various textures. The authors employ video analysis during exploration behavior and perform recordings of first-order neurons during controlled interaction between whiskers and textures placed on a rotating cylinder. Based on systematic differences in the way caudal and rostral whiskers respond to various sandpaper textures, the authors propose a coding scheme whereby caudal whiskers transmit mainly "where" information, whereas rostral shorter vibrissae transmit both "where" and "what" information.

The manuscript contains a number of interesting and novel findings: (1) There is a robust influence of the biomechanical properties of the vibrissae on the transformation of surface coarseness into whisker vibrations. (2) The texture induced vibrations are more distinct in the rostral whiskers compared to the caudal ones. (3) The neuronal responses indicate that the first-order neurons corresponding to the caudal whiskers responded mainly to edges whereas those corresponding to the rostral whiskers responded to both edges and textured stimuli.

This is a solid study that approaches a longstanding problem in rodent whisker sensory system; the role of biomechanics of whiskers in encoding of textured objects. I have a few suggestions mostly to improve the analysis and presentation of the data. 

1. There is minimal presentation of raw data:

- It would be useful to include raster plots of neuronal spiking activity as the whiskers interact with the edges and the textures. (These could be aligned to the contact with the edge, similar to the PSTHs in Figure 5A.) This could be included for a few example neurons, and would provide the reader with an idea of trial to trial variability in the neuronal response to a particular texture.

- In Figure 1, it would be nice to include an inset to show the vibrations induced by the two textures at much higher temporal resolution. 

- A clearer justification for using the normalized SD of vibrations. I am confused by the fact that the highest values of normalized SD in Figure 4C are higher than 1. This is particularly apparent for the green and red curves. 

- Similarly, for Figure 4D the use of "normalized" slope needs better justification.

- The statistical analysis related to Figure 6D presented in the form of inequality signs are not clear and difficult to understand.

2. I found the video analysis in Figure 8, the least convincing part of the manuscript. The authors argue that when the rat approaches a small food pellet, the rostral whiskers are directed forward, whereas caudal whiskers point more backward. Presumably the rat is approaching the small food pellet to sniff it or to put it in its mouth. In which case it makes sense for the most rostral whiskers to be directed forward to sense the pellet as the rat moves towards the pellet. It is difficult to conclude from this observation that the rat would use its rostral whiskers to examine any texture. For example, the situation might have been different if the rats were to examine the textured walls while running through a tunnel. 

Minor:

There are a number of typos that should be corrected by a more careful proof reading. For example:

Page 10; Second paragraph: 

The panel show that the range of ...

... the rostral whisker respond with a higher range of ...

This implies for all whiskers. 

Page 12; Second paragraph

We found a steeper slope for the rostral whisker then the caudal whisker,

Legend for Figure 1C

An heat map ...

Legend for Figure 6D

The inequality sign indicates a statistically significant differences between the various arcs.

The use of "object" and "edge" are at times confusing. For example, in Figure 1A you indicate "Edge", and in 1B, this is referred to as "Object". I think a consistent usage of "edge" would be clearer. 

Reviewer #2:

[identifies himself as Mathew E. Diamond]

Major Points

While I did appreciate the contents of the paper, the authors are overlooking one substantial form of analysis that would potentially give much more meaning to their results. The issue is introduced by this sentence:

"One major feature that stem from our results is that within a single whisk, rats may sample a region of an object at multiple spatiotemporal scales simultaneously."

As far as I could tell (and correct me if wrong, and then ignore this point) the investigators did no analysis of combined signals from multiple whiskers. I most strongly recommend doing ROC or their preferred decoding method where the observer attempts to identify the texture using the signal from 1 whisker alone, where the whisker is positioned from rostral to caudal. Now with 2 whiskers, where both whiskers have the same location (the same exact whisker twice, as a control comparison) and where the 2 whiskers have different positions. Same for 3, 4, 5 whiskers. Analyze how the decoding performance increases with number of whiskers, and test whether whisker DIVERSITY makes a difference. For instance, a rostral whisker alone may carry a better signal than a caudal whisker, but caudal plus rostral may carry a supralinearly high signal. And maybe 1 caudal whisker signal combines with 1 rostral whisker signal better than 1 rostral plus 1 rostral. In other words, diversity enriches the potential combinations in highly dimensional space and caudal whiskers might do more than supposed when looked at singly.

Additional Points

INTRO

"Facial whiskers, or vibrissae, are found in many mammalian species. They project outwards and forwards from the nose of the animal to form a tactile sensory array that surrounds the head [1]. "

1. I have heard that it is not just MANY but ALL mammals except humans, but I do not have a scientific source for that.

2. Nose: I do not think this is correct … cheeks, chin, … "snout" is a common term… not nose.

"non-moveable"

non-mobile would be a more correct term.

"Rodents use their whiskers to detect and distinguish a variety of tactile features in their environment [8] including edge position [9, 10], shape [5, 11], aperture and gap width [12], and texture discrimination [7, 13-18]."

Eliminate final word, "discrimination" - redundant with "detect and distinguish"

"An extension of this concept lies also in the "kinetic-signature" hypothesis [21, 27, 28] …"

As far as I remember, the term "kinetic signature" was not ever used in relation to a whisker's intrinsic properties, but in relation to the distinct motion caused by contact with different surfaces. I thought the signature was of the stimulus, not the whisker. I think the hypothesis of "splitting the tactile vibration signals into labeled frequency lines in the cortex" is quite different from the "kinetic signature" hypothesis. I may be wrong but I would like the authors to re-check.

METHODS

"Each revolution of the wheel lasted for about 3 sec. and no noticeable adaptation in neuronal firing rates were detected."

Please comment on whether this disagrees with Hartmann claim of some adapting neurons in the ganglion. In fact, later (Results) we have: "These firing patterns are conventionally used to characterize neurons as either rapidly adapting (RA) or slowly adapting (SA), respectively. We did not find any significant bias in neuronal types towards any of the arcs (Table 1)."

"To examine the influence of whisker identity on responses to edges and textures (Fig. 1A,B), we separated edge and texture related stimulation epochs by identifying big excursion (mean±3SD) in whisker vibrations during wheel rotation."

Is this a circular argument? I mean, are the authors deciding a priori what form of excursion whiskers must use in order to encode textures and edges?

RESULTS

"Vibrissae movements were measured in response to sandpapers having two different grades: P220, P400. However, since we got similar results for all textures (not shown), we combined all these results together."

This does not make sense at all, to me. Combined what? Please clarify the sentence.

Please give more informative labels to Figure 1C-D.

 DISCUSSION

"In this scheme, longer caudal whiskers transmit mainly "where" information, whereas rostral shorter vibrissae transmit both "where" and "what" information."

This is a commonly made statement, going back to Brecht et al. 1997 and it may be valid when rats have full access to the object of interest with all whiskers. But rats are frequently "reachers," e.g. extending themselves from one step to feel the one below on the stairway down to the subway station. When they cannot contact with micros, they may very well use macros for texture and other fine judgments. In fact many behavioral experiments show excellent texture discriminations with macros, when the rat can reach only with those.

The paragraph beginning

"We found that whiskers, which are the first stage of tactile information translation, form a location- dependent gradient in which rostral whiskers show smaller attenuation in the transformation from texture coarseness to whisker vibrations…"

is a mini-review of texture coding by the whiskers, and in that context the authors will be interested in a recent paper, Zuo et al. CurrBiol "Rats Generate Vibrissal Sensory Evidence until Boundary Crossing Triggers a Decision" which breaks down some of the whisker kinematic features that distinguish texture.

There are several misspellings and typos throughout the paper, e.g. "criterions". I have not listed them.

Reviewer #3:

This paper explores a relatively well-trod path of investigation into whisker characteristics in texture and edge detection. The vibration analysis (Fig 1-3) and whisker behavior analysis (Fig 8) is rudimentary. The former is useful more as a refresher on some whisker characteristics, and the latter does not push far enough to increase our understanding of foraging behavior. However, the middle of the paper with the trigeminal ganglion recordings of texture vs. edge responses are quite interesting. In particular, Figure 5 shows a beautiful dissociation of edge versus texture response across the pad. The impact of this on discrimination is made clear in Figs 6 & 7. The overall concept of a gradient of specialization for edge vs. texture discrimination appears sound and well supported. Moreover, this is a pretty compelling distillation of an organizing principle. Thus, at the core, this paper has a lot of potential. 

I believe this paper would have greater impact if the less groundbreaking (1-3) or less developed (8) parts of it were condensed into fewer figures or eliminated. A more compact presentation of only the key data pieces needed to support the main punchline of discrimination gradient, plus the existing detailed characterization of the TG gradient would bring the strengths of the work to the forefront. 

Detailed comments below:

Figs 1 & 2. The shorter whiskers had higher positional variance (SD) and deflection from the edge during the rotation of the texture wheel. Bigger whiskers have more inertia, so the same level of force might be expected to move them less. Yet bigger whiskers also sweep across a wider range of motion, which could exert more force on objects during active whisking. Would increased force applied by the bigger whiskers during whisking be sufficient to normalized the level of vibration across the grid?

Figure 2 varies applied force by changing distance of wheel from the face and shows the same gradient of sensitivity exists, but this doesn't quite address the question, as the force change is uncontrolled for each whisker. How was the force applied by the wheel calibrated across the grid? These results would be strengthened by a more rigorous calculation of lateral force and bending moment applied for each whisker position. The overall lateral force and bending moment could likely be computed from known whisker mechanical properties (see work by Hartmann, Svoboda, Golomb) and the degree of curvature change from videos that have already been collected.

Figure 3 makes a fairly sweeping claim that "Mechanical characteristics of different whiskers rows are not different from each other." Even with the text results limiting this to "responses to textures" it seems that this claim is more a reflection of the coarse measurement approach (SD) rather than the whiskers themselves. Certainly whiskers that have varying lengths or diameters will have different vibrational characteristics that are not captured in this one-dimensional characterization (e.g. classic work by Chris Moore, or more recent computational studies by Shulz, Debrégeas and Wandersman, or Hires and Efros). However, SD may be a sufficient level of characterization for firing rate base texture discrimination models used in the TG recording analysis.

Fig 4. I'm not sure that using different scale bars in A is worth it. While blowing up gamma lets you see the variance a bit better, the other key point authors are making is in the differences in vibration amplitude. It would be more visually compelling with gamma having the same scale bar as C4. Furthermore, the single trial data does not appear to match 4B. After normalizing scale bars, gamma looks about half the SD as C4 in A, but looks <0.1 the SD of C4 in B. What is the reason for that discrepancy? Also, why is B y-axis starting at 0.05 instead of 0? Fig 4C-E are more clear and compelling. Here they show that the small whiskers more faithfully discriminate grain size with the SD variable. This is a key finding for the paper.

Fig 5 This is the key figure in the paper and it is a quite striking result. The longer whiskers clearly are insensitive to texture but retain robust edge responses. On the other hand, the edge responses are hard to pick out from texture responses in the short whiskers. 5A is a beautiful visualization of this gradient. Now the benefit of the experimental apparatus with both texture and edge stimuli is clear.

Fig 6&7 Are a nice extension of the results in 5 showing the discrimination gradients are arc not row dependent. Fig 7 is somewhat repetitive with Fig 6. Is there a way to demonstrate the robustness with fewer panels?

Fig 8. This figure adds little and should either be significantly improved or eliminated. The repetition of all 6 mice that basically shows the same thing serves little purpose. It's obvious that caudal whiskers point more backward then rostal ones, that's how the follicle array is designed. Is there something special that happens when the animal is foraging that isn't apparent when looking at a rat at rest? It's hard to understand how the analysis is being performed. It's confusing to use SD as a metric for whisking angle when it was previously used for vibration range. Why not use field standard approaches to measuring amplitude with a Hilbert transform? At least label the y-axis something like "Whisking amplitude (SD)" Some annotated single trial whisker angle time series data between A and the rest of the figure would help guide the reader.

---

## [Decision Letter · Decision Letter 2]

3 Aug 2020

Dear Rony,

Thank you for submitting your revised Research Article entitled "Gradient of Tactile Properties in the Rat Whisker Pad" for publication in PLOS Biology. I have now obtained advice from the original reviewers and have discussed their comments with the Academic Editor. 

Based on the reviews, we will probably accept this manuscript for publication, assuming that you will modify the manuscript to address the remaining points raised by reviewer #3. Please also make sure to address the data and other policy-related requests noted at the end of this email.

Overall, the reviewers are very positive regarding your study, and have each recommended its acceptance for publication at PLOS Biology (you can find their detailed comments at the end of this letter). IMPORTANT:

a) Reviewer 3 has asked for several minor revisions including the addition of display images and additional labeling within figures/adjustment of figure legend text to improve readability. We therefore would like to invite you to please address reviewer 3’s comments. 

b) Please include within your manuscript the name of the specific IACUC that reviewed the study protocol, the name of the specific international ethical guidelines to which your approved protocols adhered, and the ID number of the protocol approved by the IACUC.

c) Please address my Data Policy requests (see further down).

We expect to receive your revised manuscript within two weeks. Your revisions should address the specific points made by each reviewer. In addition to the remaining revisions and before we will be able to formally accept your manuscript and consider it "in press", we also need to ensure that your article conforms to our guidelines. A member of our team will be in touch shortly with a set of requests. As we can't proceed until these requirements are met, your swift response will help prevent delays to publication.

*Copyediting*

*Published Peer Review History*

*Early Version*

*Submitting Your Revision*

Sincerely,

Roli

Senior Editor,

rroberts@plos.org,

PLOS Biology

ETHICS STATEMENT:

-- Please include the full name of the IACUC/ethics committee that reviewed and approved the animal care and use protocol/permit/project license. Please also include an approval number.

-- Please include the specific national or international regulations/guidelines to which your animal care and use protocol adhered. Please note that institutional or accreditation organization guidelines (such as AAALAC) do not meet this requirement.

-- Please include information about the form of consent (written/oral) given for research involving human participants. All research involving human participants must have been approved by the authors' Institutional Review Board (IRB) or an equivalent committee, and all clinical investigation must have been conducted according to the principles expressed in the Declaration of Helsinki.

DATA POLICY:

Regardless of the method selected, please ensure that you provide the individual numerical values that underlie the summary data displayed in the following figure panels as they are essential for readers to assess your analysis and to reproduce it: Figs 1B, 2BDF, 3ABCDEF, 4ABCDEF, 5BCDE, 6BC, 7ABCD, S1AB, S2B, S3ABCD. NOTE: the numerical data provided should include all replicates AND the way in which the plotted mean and errors were derived (it should not present only the mean/average values).

REVIEWERS' COMMENTS:

Reviewer #1:

[identifies himself as Ehsan Arabzadeh]

 The authors have addressed the issues raised in my earlier review. I have no further comments. 

Reviewer #2: The revisions are thorough and the study looks publishable.

Reviewer #3:

The Fig 1B traces are a welcome addition. 

Figure 2 & 3 look good.

The new figures on supralinear summation are interesting. The AUC data in Figure 4 would benefit from the display of an example ROC curve for one whisker. The multiwhisker analysis would benefit from display of individual ROC curves for two whiskers overlaid with the ROC curve for their combined discrimination power.

My remaining suggestions relate to style.

Some figures are still hard to understand at a glance. This could be alleviated with judicious labelling within the figures. For example, 1B could be labelled "Whisker vibrations", 1C could be be labelled "Texture response." in 1D the label "Normalized SD" is more a designation of units, not of the feature being measured (Whisker vibrations). The authors could improve the readability of their manuscript by examining each figure subpanel and giving them the best, succinct label. Figure 2 is better in this regard, but many other figures plots are not optimally labelled. 

Likewise, all of the figure titles could be improved. I recommend making the figure title be an accurate take home message of the figure, rather than a restating of the plots. For example, instead of "Response sensitivity of the different whisker arcs to different textures", a better title would be "Shorter whiskers exhibit higher response sensitivity to texture grain". Priming the reader with titles in that style for all figures will help them more quickly grasp the result. 

Intro:

I recommend the following tweaks

" thereby differentially influencing "

" of each individual whiskers "

---

## [Editor Report · Decision Letter 3]

14 Sep 2020

Dear Dr Azouz,

On behalf of my colleagues and the Academic Editor, Carl C.H. Petersen, I am pleased to inform you that we will be delighted to publish your Research Article in PLOS Biology. 

Early Version

PRESS 

Kind regards,

Vita Usova

Publication Assistant, 

PLOS Biology

on behalf of

Roland Roberts,

Senior Editor

PLOS Biology